# AKR1B10, One of the Triggers of Cytokine Storm in SARS-CoV2 Severe Acute Respiratory Syndrome

**DOI:** 10.3390/ijms23031911

**Published:** 2022-02-08

**Authors:** Clovis Chabert, Anne-Laure Vitte, Domenico Iuso, Florent Chuffart, Candice Trocme, Marlyse Buisson, Pascal Poignard, Benjamin Lardinois, Régis Debois, Sophie Rousseaux, Jean-Louis Pepin, Jean-Benoit Martinot, Saadi Khochbin

**Affiliations:** 1Institute for Advanced Biosciences—UGA—INSERM U1209—CNRS UMR 5309, 38700 La Tronche, France; anne-laure.vitte@univ-grenoble-alpes.fr (A.-L.V.); domenico.iuso@univ-grenoble-alpes.fr (D.I.); florent.chuffart@univ-grenoble-alpes.fr (F.C.); sophie.rousseaux@univ-grenoble-alpes.fr (S.R.); saadi.khochbin@univ-grenoble-alpes.fr (S.K.); 2Laboratoire BEP (Biochimie des Enzymes et les Protéines), Institut de Biologie et de Pathologie, CHU Grenoble Alpes, 38700 La Tronche, France; ctrocme@chu-grenoble.fr; 3Institut de Biologie Structurale, CEA, CNRS and Centre Hospitalier Universitaire Grenoble Alpes, Université Grenoble Alpes, 38000 Grenoble, France; marlyse.buisson@ibs.fr (M.B.); pascal.poignard@ibs.fr (P.P.); 4Laboratory Department, CHU UCL Namur Site de Ste Elisabeth, 5000 Namur, Belgium; benjamin.lardinois1@gmail.com (B.L.); regis.debois@uclouvain.be (R.D.); 5HP2 Laboratory, INSERM U1300, Grenoble Alpes University, 38000 Grenoble, France; JPepin@chu-grenoble.fr; 6Sleep Laboratory, Pole Thorax et Vaisseaux, Grenoble Alpes University Hospital, 38000 Grenoble, France; 7Sleep Laboratory and Pulmonology and Allergy Department—CHU UCL Namur, St. Elisabeth Site, 5000 Namur, Belgium; martinot.j@respisom.be; 8Institute of Experimental and Clinical Research, UCL Bruxelles Woluwe, 1200 Brussels, Belgium

**Keywords:** cytokines, inflammation, COVID-19, drug treatment

## Abstract

Preventing the cytokine storm observed in COVID-19 is a crucial goal for reducing the occurrence of severe acute respiratory failure and improving outcomes. Here, we identify Aldo-Keto Reductase 1B10 (AKR1B10) as a key enzyme involved in the expression of pro-inflammatory cytokines. The analysis of transcriptomic data from lung samples of patients who died from COVID-19 demonstrates an increased expression of the gene encoding AKR1B10. Measurements of the AKR1B10 protein in sera from hospitalised COVID-19 patients suggests a significant link between AKR1B10 levels and the severity of the disease. In macrophages and lung cells, the over-expression of AKR1B10 induces the expression of the pro-inflammatory cytokines Interleukin-6 (*IL-6)*, Interleukin-1β (*IL-1β*) and Tumor Necrosis Factor a (*TNF**α*), supporting the biological plausibility of an AKR1B10 involvement in the COVID-19-related cytokine storm. When macrophages were stressed by lipopolysaccharides (LPS) exposure and treated by Zopolrestat, an AKR1B10 inhibitor, the LPS-induced production of *IL-6*, *IL-1β*, and *TNFα* is significantly reduced, reinforcing the hypothesis that the pro-inflammatory expression of cytokines is AKR1B10-dependant. Finally, we also show that AKR1B10 can be secreted and transferred via extracellular vesicles between different cell types, suggesting that this protein may also contribute to the multi-organ systemic impact of COVID-19. These experiments highlight a relationship between AKR1B10 production and severe forms of COVID-19. Our data indicate that AKR1B10 participates in the activation of cytokines production and suggest that modulation of AKR1B10 activity might be an actionable pharmacological target in COVID-19 management.

## 1. Introduction

The Coronavirus Disease 2019 (COVID-19) infection, which rapidly spread worldwide, was declared a pandemic on 11 March 2020 by the World Health Organisation [1]. The intense campaign of vaccinations that protect against severe COVID-19 forms in several countries currently contributes to the fight against the pandemic [2]. Nonetheless, the vaccination appears insufficient to completely prevent the circulation of the virus, and the occurrence of severe forms requiring the Intensive Care Unit (ICU) is still associated with high mortality rates [3]. Hence, developing new approaches to detect SARS-CoV2 infection [4,5] and preventing the occurrence of the most severe cases of Acute Respiratory Distress Syndrome (ARDS) associated with severe COVID-19 forms represents a crucial strategy to reduce the burden of the COVID-19 pandemic. Although COVID-19 is associated with a wide range of symptoms [6], life-threatening ARDS are tightly triggered by a massive inflammatory burst, named a cytokine storm [7]. Among the released cytokines, the Tumour Necrosis Factor α (TNFα), Interleukine-6 (IL-6), Interleukine-1β (IL-1β), or the Interferon γ (IFNγ) have been identified by several studies as major ARDS-inducing factors [7,8]. Several therapeutic strategies have tried to reduce the cytokine storm burden with a limited impact (e.g., Remdesivir [9], Ivermectin [10], hydroxychloroquine [11]). Currently, corticosteroids are the most effective anti-inflammatory drugs, reducing the mortality of ventilated patients by one-third, and of patients requiring oxygen therapy by one-fifth [12]. Nonetheless, long term use of corticosteroids during Severe Acute Respiratory Syndrome CoronaVirus 2 (SARS-CoV2) infection may also be deleterious [13]. For all these reasons, dissecting the mechanisms triggering the cytokine storm and identifying new pharmacological targets remains a burning question in COVID-19 management.

The aldo/keto reductases (AKR) are part of a super-family of NADPH-dependent enzymes (15 members in humans) which were first identified as catalysers of redox transformations, mainly involved in biosynthesis, metabolism, and detoxification [14]. In the year 2000, AKR1B10 was identified as a regulator of inflammation [15] and several publications have reported that it is required for the nuclear translocation of the Nuclear Factor kappa B (NF-κB) and phosphorylation/degradation of IκB-α, stimulating the expression of pro-inflammatory cytokines. This pro-inflammatory role of AKR1B10 makes it a potential drug target in the medical care of many chronic diseases associated with inflammation [16,17]. Inhibitors of AKR1B10 have successfully been tested in animal and cellular models to block the inflammation response induced by various stresses, including hyperglycaemia, Lipopolysaccharides (LPS), TNFα, and oxidative stress [18,19,20,21,22]. AKR1B10 is upregulated in several chronic diseases associated with low-grade inflammation, known to increase the risk of severe forms of COVID-19. Indeed, Chronic Obstructive Pulmonary Disease (COPD) [17], various types of cancer [16], diabetes [23], and Non-Alcoholic Fatty Liver Disease (NAFLD) [24,25], strongly associated with type II diabetes [26,27], are conditions associated with both high AKR1B10 expression and a high proportion of severe COVID-19 forms.

Here, an analysis of transcriptomic data from COVID-19 patients’ post-mortem lung tissues first confirmed that AKR1B10 could be a potential contributor to the occurrence of severe respiratory failure in COVID-19 patients. Following this hypothesis, we then designed a study to evaluate AKR1B10 as a potential mediator of the cytokine storm in COVID-19 patients, as well as to test the ability of AKR inhibitors to reduce the inflammatory burst which occurs in the severe forms.

## 2. Results

### 2.1. AKR1B10 Is Overexpressed in the Lung of Deceased COVID-19 Patients and Correlated with an Enrichment of Pro-Inflammatory and Cytokine Genes

The analysis of transcriptomic data of lung samples from patients who died from COVID-19 shows a significant increase of *AKR1B10* (Fold Change (FC): 8.58; adjusted *p*-value: 1.07 × 10^−6^), ranking it as the 13th most over-expressed gene (Figure 1A).

Interestingly, *AKR1B10* is the only overexpressed gene of the *AKR1B* gene family, since the over-expressions of *AKR1B1* and *AKR1B15* do not reach significance (FDR over the threshold value fixed at 0.02). A GeneSet Enrichment Analysis (GSEA, [29]) of the genes differentially expressed in the post-mortem lungs of COVID-19 patients (Figure 1B) highlights a major role for an inflammatory process involving the over-expression of cytokines or their receptors. Indeed, the genesets associated with the Gene Ontology terms “cytokine production” (NES: 6.18; FDR < 0.001), as well as the “positive regulation of inflammatory response” (NES: 7.79; *p* < 0.001), were found among the most significantly enriched genesets.

### 2.2. AKR1B10 Levels Are Increased in the Blood of COVID-19 Patients with Severe or Critical Forms of the Disease Compared to Moderate Forms

AKR1B10 presents the particularity of circulating in the human blood stream upon its expression. Indeed, a previously published work reported that AKR1B10 could be detected in the blood of patients suffering from different forms of cancers [30,31]. Based on these data, we hypothesised that sera from COVID-19 patients could be used to evaluate the extent of AKR1B10 expression and to link sera levels to the severity of the disease. Consequently, a dosage of AKR1B10 was performed in the blood samples from a cohort of 104 patients, and the results were analysed considering their disease severity and various biological and physiological parameters (detailed in Table 1).

As expected, a comparison between the groups of patients respectively admitted in an intensive care unit (ICU) or a non-ICU respiratory ward shows a significant difference in survival rates (respectively, 37.2% and 98.4%; *p* < 0.001), a lower number of ICU patients without any comorbidity (−29%, *p* < 0.01) and a higher number of ICU patients with one comorbidity (+22%, *p* < 0.05). The physiological and biological data show that the patients with a severe COVID-19 form have lower PaO_2_ measurements (measured without supplemental O_2_ at hospital admission, −6.6 mmHg; *p* < 0.01), decreased lymphocyte counts (−0.71 abs. ×10/µL; *p* < 0.001), higher Ferritin concentrations (+371.1 ng/mL, *p* < 0.05), and higher CT-Scan scores (+32%; *p* < 0.001, based on the lung areas showing ground glass opacity and patchy consolidations). All these parameters are known to be associated with COVID-19 severity and poor outcomes.

The AKR1B10 protein concentrations measured in the sera of COVID-19 patients presented in Figure 2 shows a relationship between the concentration of the protein and the severity of the disease.

Indeed, the patients included in the study, who developed a COVID-19 disease requiring hospitalisation in a non-ICU respiratory ward or in an ICU, have significantly higher levels of AKR1B10 when compared to non-COVID control subjects (respectively, +2.8 ng·µL^−1^, *p* < 0.05; and +4.3 ng·µL^−1^, *p* < 0.001). Patients developing a severe form of the disease (ICU) have higher concentrations of AKR1B10 when compared to non-ICU patients (+1.5 ng·µL^−1^, *p* < 0.01), and all patients with a concentration over 8.1 ng·µL^−1^ were hospitalised in the ICU. These data strongly suggest a positive correlation between AKR1B10 levels and the severity of COVID-19. The hypothesis of this association is further reinforced by comparing the respective balanced proportions of the non-COVID, non-ICU and ICU COVID patients within each of the four quartiles, Q1 (lowest) to Q4 (highest), of AKR1B10 concentrations in sera (Figure 2B). Indeed, the proportion of non-COVID subjects decreases from 75% in Q1 to 0% in Q4, whereas the proportion of ICU patients increases from 6% in Q1 to 66% in Q4. No significant change is detected for non-ICU patients whose percentage is maintained around 37 ± 0.07% between quartiles.

An analysis of the correlations between AKR1B10 concentrations in the sera of COVID-19 patients and other parameters known to be associated with inflammation and available for this cohort highlights two significant associations. Indeed, AKR1B10 concentration is negatively correlated with the lymphocyte counts (R²: 0.38; *p* < 0.001; Figure 2B) and positively correlated with the lactate dehydrogenase (LDH) levels (R²: 0.30; *p* < 0.01; Figure 2B). However, no significant correlation with the C-reactive protein (CRP) concentration is observed (R²: 0.06; ns; Figure 2B), although CRP is one of the most widely used blood markers of inflammation.

The analyses of lung transcriptomic data and blood sera from COVID-19 patients, respectively measuring AKR1B10 mRNA and protein expression, highlight a potential relationship between this factor and the severity of COVID-19. Since most severe forms of COVID-19 are associated to an intense production of cytokines related to the pro-inflammatory process (see Figure 1B), we used cell models to modulate the AKR1B10 expression or activity, in order to test its involvement in the increased expression of cytokines.

### 2.3. AKR1B10 Drives Cytokine Production in Cellular Models

To investigate the role of AKR1B10 on inflammatory processes and particularly on the induction of the cytokine storm, we overexpressed AKR1B10_GFP_ in a murine macrophage cancerous cell line and in a human non-small cell lung carcinoma cell line (respectively, RAW264.7 and H1299) and tested the inflammatory response of the cells by monitoring the mRNA expression of three different cytokines (*IL-6*, *TNFα* and *IL-1β*) known to be involved in the cytokine storm. Several inhibitors of the AKR1B10 enzyme activity already exist and have been successfully tested in different contexts of inflammation. We chose one of them, Zopolrestat, and tested its effect at different concentrations on RAW264.7 cells in the context of a 0.5 µg·mL^−1^ lipopolysaccharides (LPS)-induced stress (Figure 3).

Figure 3A shows a clear induction of cytokines mRNA expression by AKR1B10_GFP_ overexpression in RAW264.7 macrophages. Indeed, when compared to the control group the transfection with 3 µg of pEGFP-AKR1B10_GFP_ plasmid induces a 54-fold change of *IL-6* (*p* < 0.001), a 12-fold change of *TNFα* (*p* < 0.001), and a 30-fold change of *IL-1β* (*p* < 0.001). Interestingly, these results suggest a dose–response relationship between AKR1B10_GFP_ plasmid concentration and the cytokines production for *IL-6* at 1 µg versus 3 µg (respectively, 33- and 54-fold change from the control; *p* < 0.05) and *TNFα* at 1 µg and 2 µg versus 3 µg (respectively, 7-, 8-, and 12-fold change from control, *p* < 0.05). Over expression of AKR1B10_GFP_ [1 µg] in H1299 lung cancer cells (Figure 3B) induces similar effects but with a lower order of magnitude for *IL-6* and *IL-1β* (respectively, 3-fold change from control, *p* < 0.01 and 1.5-fold change from control, *p* < 0.05) than those observed in macrophages, and no significant modification of *TNFα* expression (1.1-fold change from control, ns). These results suggest a key role for AKR1B10 in the pro-inflammatory cytokines production, which could make this factor an interesting therapeutic target to prevent or reduce the cytokine storm observed in a COVID-19 context.

In order to further explore this hypothesis, we treated RAW264.7 cells, after 6 h exposure to a well-known pro-inflammatory inducer, the lipopolysaccharides (LPS), with different concentrations of Zopolrestat to inhibit the AKR1B10 activity (Figure 3C). The mRNA measurements of the cytokines in macrophages stimulated by LPS show an increase in *TNFα* and *IL-1β* expression, which is significantly reduced by 40 mM of Zopolrestat when compared to untreated cells (respectively, −25%; *p* < 0.05 and −36%; *p* < 0.01). With a higher concentration of Zopolrestat (80 mM), the mRNA expression of all three cytokines is significantly decreased when compared to LPS exposure without Zopolrestat (*IL-6*: −27% *p* < 0.05; *TNFα*: −36% *p* < 0.01 and *IL-1β*: −53% *p* < 0.001).

### 2.4. The AKR1B10 Protein Can Be Transferred between Cell Types via the Extracellular Vesicles

AKR1B10 is known to be secreted via the Extracellular Vesicles (EVs) [32] and has been identified as an interactor that favours metastasis growth in an indirect co-culture cancer model [30]. We therefore hypothesised that the AKR1B10 protein could be transferred between different cells types via the EVs pathway, a process which may play a role in the systemic repercussion of COVID-19 (Figure 4).

In order to test this hypothesis, EVs were extracted from H1299 cells expressing different levels of AKR1B10_GFP_ (Low to High, see Supplementary Appendix AA for details), and levels of AKR1B10 in these EVs were estimated by Western blots (Figure 4A). Extraction by ultracentrifugation allowed for the sorting of EVs into two sub populations, the large EV, mainly composed of apoptotic bodies and microvesicles, and the Exosomes corresponding to small EVs with a diameter under 150 nm [33]. In order to test the possibility of the AKR1B10_GFP_ protein to be transferred between cell types, we exposed non-transfected RAW264.7 cells to EVs from H1299 cells, either expressing high levels of AKR1B10_GFP_ (H1299_High_) or without AKR1B10_GFP_ transfection (H1299_Ct_), and measured the GFP signal by FACS (Figure 4B).

Figure 4A shows a clear relationship between the intracellular level of AKR1B10_GFP_ and its concentration in the large EVs and exosomes. Indeed, a significant correlation between AKR1B10_GFP_ contained in cells and EVs is observed (Supplementary Appendix AB; R²: 0.966; *p* < 0.001). The density curve of the GFP signals of RAW264.7 cells exposed to EVs containing AKR1B10_GFP_ proteins shows a shift to the right (Figure 4B, FL1-H), reflecting an increase of cells with a higher GFP emission suggesting an AKR1B10_GFP_ uptake. This result is also supported by a significant increase in the mean fluorescence intensity of the FL1-H signal (*p* < 0.001; Figure 4C). These experimental data come in support of the hypothesis that AKR1B10 can be secreted and transferred between cell types. Together with its presence in the blood of patients with severe COVID-19 this observation suggests that AKR1B10 could also be involved in the systemic inflammatory syndrome associated with the disease.

## 3. Discussion

This study, based on data obtained from lung and blood samples of COVID-19 patients, highlights a positive association between AKR1B10 expression and the severity of the disease. Mechanistic investigations in cellular models suggest a tight relationship between AKR1B10 expression and the induction of pro-inflammatory proteins known to be associated with the cytokine storm, which may be counteracted by inhibiting AKR1B10′s catalytic activity. Therefore, our results suggest that an increased level of AKR1B10 in the lung of COVID-19 patients may participate in the production of cytokines associated with the ARDS, an observation which could open promising perspectives to reinforce the therapeutic arsenal for the COVID-19 disease.

One of the main outcomes of this work is the demonstration of increased ARK1B10 levels in the blood samples of patients with severe COVID-19 disease, based on their admission in ICU, compared with that of other COVID-19 patients (Figure 2A,B).

AKR1B10 blood concentration also significantly correlates with lymphopenia and LDH levels, parameters which have already been associated with the cytokine storm [34,35,36]. Although CRP concentration is also a well-known and widely used biomarker of inflammation, it does not correlate with our measurement of AKR1B10 in sera, an observation which may be explained by the high levels of CRP observed in all the patients included in the study.

The analysis of the overexpressed genes in post-mortem COVID-19 patients’ lungs suggest that the over-expression of AKR1B10 is associated with the high enrichment of genes involved in the inflammation response and in the production of cytokines. In line with this observation, AKR1B10 has been identified as a key protein in the activation of the cytokines’ production in different models and inflammatory contexts [18,19,20,21,22]. Thus, the increased expression of *TNFα*, *IL-6*, and *IL-1β*, observed here in the macrophage cells RAW264.7 or in the lung cancer cells H1299 (Figure 3), overexpressing AKR1B10 or being stressed by LPS (Figure 3C), confirms a critical role for this enzyme in mediating cytokine gene expression, which is in line with the current literature [18,19,20,21,22]. Interestingly, in cells exposed to LPS, despite the absence of an LPS-induced increase of AKR1B10 concentration that is in line with the current literature [37], we observe an increase of cytokines’ expression (Cf. Supplementary Appendix AB). This result suggests that basal enzymatic activity of the protein is sufficient to participate in the inflammatory process during cellular stress.

Interestingly, a study by Wang et al. published in 2007 [38] during the SARS-CoV1 epidemics, showed that exposure of RAW264.7 cells to the SARS-CoV1 spike protein resulted in an increased production of cytokines, suggesting that AKR1B10 inhibitors could be used as a potential therapeutic approach in this context. Another study focused on the pharmacological targets of the ShenFuHuang formula, a traditional Chinese medicine widely used in association with the clinical treatment of COVID-19 in Wuhan, identifies 69 proteins including AKR1B10, with a possible role in COVID-19 [39]. A recent study ([40], pre-print) demonstrates that patients treated with Fenofibrate show a rapid reduction in inflammation and a significantly faster recovery compared to control patients admitted during the same period and treated with the standard-of-care. Interestingly, our results suggest that this effect may actually be mediated by the AKRs inhibiting activity of the Fenofibrate [41]. Following this hypothesis, AKR inhibitors could represent a promising therapeutic leverage in COVID-19 disease. Nonetheless, there are multiple inhibitors of AKR1B10 activity with different levels of efficiency, and clinical trials would be required as a key step to define the most promising molecules and to improve the positive effects already observed with Fenofibrate.

AKR1B10 is known to be secreted via a non-canonical pathway due to its interaction with the HSP90 protein [32], and could act distantly since it favours metastasis growth in an indirect co-culture cancer model [30]. Our data in the lung cancer cell line H1299 support the hypothesis that ARK1B10 exocytosis in large and small extracellular vesicles could be associated with its expression. Furthermore, our data also suggest that a transfer of AKR1B10 protein between different cell types is possible via the EVs (Cf. Figure 4). Therefore, during COVID-19, it is possible that a high blood concentration of AKR1B10 is a consequence of an increased expression of AKR1B10 in the lung, and that the AKR1B10 transport through the blood stream could favour the production of pro-inflammatory cytokines in various organs other than the lung, which could participate to the multi-systemic effects associated with COVID-19 related ARDS. The study of Daamen et al. [28] which shows no difference in the mRNA transcription of AKR1B10 in the peripheral blood monocytes of COVID-19 patients also supports this hypothesis.

### Strength and Weakness

To our knowledge, this study is the first focused on the links between the AKR1B10 protein and the cytokine storm that occurs in severe forms of COVID-19. Indeed, our data support a key role for AKR1B10 in sustaining ARDS following SARS-CoV2 infection. The pro-inflammatory role of AKR1B10 was already known, but this possibility had never been considered in the context of COVID-19 associated ARDS. More investigations are needed to monitor the AKR1B10 expression kinetics during COVID-19 disease progression in order to determine if the high AKR1B10 expression detected in the lungs and blood of patients is also observed prior to the viral infection, in which case its dosage in the sera could be a risk predictor of severe COVID-19 forms. Another limitation of the study is the absence of cytokines’ over-expression by RAW264.7 cells when exposed to EVs. However, in our system, the levels of AKR1B10 measured in the cell medium (0.37 ± 0.14 ng·µL^−1^; Cf. Supplementary Appendix AA), are much lower than the concentrations measured in the sera of the COVID-19 patients, even after EVs’ extraction. Therefore, although this experimental setting is sensitive enough to demonstrate the presence of AKR1B10 vesicles capable of fusing to other cells, it is not appropriate to test the effect of AKR1B10 on the expression of cytokines. An important point raised by this study is the possibility to use a new family of chemical AKR1B10 inhibitors to counteract the cytokines’ production that occurs during the COVID-19 induced cytokine storm. Further studies are required to test the benefit of AKR inhibitors in a clinical trial context.

## 4. Materials and Methods

### 4.1. Sera

The sera of patients (*n* = 110) were obtained from the biobank of the CHU UCL Namur site Godinne, notified to the Belgian federal agency of medicines and health products (AFMS) under the reference number: BB190023. Two patients who were under kidney dialysis were removed from the study. Three groups were performed: (i) “Non-COVID” (*n* = 16); (ii) “non-ICU” (*n* = 61) corresponding to patients admitted in a non-ICU respiratory ward; (iii) “ICU” (*n* = 43) corresponding to patients hospitalised in an Intensive Care Unit. AKR1B10 protein measurements in the sera were performed with a 241SEQ201Hu kit (Cloud-Clone Corp., Houston, TX, USA) according to the manufacturer’s instructions. All data were collected and analysed in accordance with the ethics committee recommendations and the Helsinki Convention.

### 4.2. Cell Culture

RAW264.1 and H1299 cells were obtained from the American Type Culture Collection company (ATCC, Manassas, Va., USA) and cultured in Gibco RPMI1640 + Glutamax media (Thermo Fisher Scientific, Waltham, MA, USA) with 10% of foetal bovine serum (Corning Life Sciences, Corning, NY, USA)and antibiotics PenStrep 1X (Thermo Fisher Scientific, Waltham, MA, USA). All the transfections were performed with lipofectamine2000 (Thermo Fisher Scientific, Waltham, MA, USA), Optimem (Thermo Fisher Scientific, Waltham, MA, USA), and a pEGFP-C2 vector containing the coding sequence of AKR1B10. For the transient AKR1B10_GFP_ expression, RAW264.7 and H1299 cells were transfected 12 h before mRNA extraction in 6 wells plate with 200 µL of Optimem, 1–3 µg of plasmid and 3–9 µL of Lipofectamine2000. In order to induce a permanent expression in H1299, transfection was performed in 1000 mm well plates with 500 µL of Optimem, 5 µg of plasmid, and 15 µL of Lipofectamine2000, and the cells were cultured for 2 weeks with 100 µL of Neomycin (Thermo Fisher Scientific, Waltham, MA, USA) to remove non-transfected cells and then sorted with a cell sorter (Aria IIu Cell Sorting System, Becton Dickinson, Franklin Lakes, NJ, USA) according to the terciles of GFP expression levels, corresponding to H1299_Low_, H1299_Medium_, and H1299_High._

### 4.3. RT-PCR

RNA from RAW264.7 cells were extracted with Nucleospin mini-columns (Macherey-Nagel Inc., Allentown, PA, USA) and reverse transcribed with RT affinity script (Agilent Technologies, Santa-Clara, CA, USA). The cDNA and no Reverse Transcript (NoRT) samples were diluted in 1/20 before performing real-time PCR with the Brilliant III kit (Agilent Technologies, Santa-Clara, CA, USA) in Stratagene Mx3000P RT-PCR machine (Agilent Technologies, Santa-Clara, CA, USA). All data were quantified according to the ΔΔCt method [42], with normalisation using two housekeeping genes (GAPDH and Actin). The sequence primers are available in Appendix A.

### 4.4. Western Blotting

Electrophoresis was performed on a Mini-Protean system (Bio-Rad Laboratories, Hercules, CA, USA) with precast NuPAGE™ gradient gel 8–20% (Thermo Fisher Scientific, Waltham, MA, USA) followed by transfer on a nitrocellulose membrane 0.45 µm Protran (Amersham^TM^, Amersham, UK) using the FastTurbo blot machine (Bio-Rad Laboratories, Hercules, CA, USA). The antibodies used for quantification were AKR1B10 (Ab96417, Abcam, Cambridge, UK) and Flotillin-1 (AB_398139, Becton Dickinson, Franklin Lakes, NJ, USA).

### 4.5. Extracellular Vesicles (EVs) Extraction and Exposure

H1299 cells were cultured to 80% confluence in 250 mm plate, then washed once in PBS 1X, and cultured in Gibco RMPI1640 + Glutamax (Thermo Fisher Scientific, Waltham, MA, USA) medium serum and antibiotics free for 24 h to favour Extracellular Vesicles (EVs) secretion. Large EVs and exosomes were extracted by ultra-centrifugations (respectively, 10,000× *g* for 30 min and 100,000× *g* for 70 min) in an Optima LE-80K with a SW28 rotor (Beckman Coulter, Brea, CA, USA) and re-suspended in 50 µL of PBS 1X according to a previously described protocol [43].

Non transfected RAW264.7 cells seeded in 12 wells plate were exposed to EVs extracted from H1299_Ct_ and H1299_High_ cells for 12 h. After the incubation period, cells were washed with PBS 1x, trypsinised (Thermo Fisher Scientific, Waltham, MA, USA) and the Green signal (FL1) was analysed in an Accury C6 (Becton Dickinson, Franklin Lakes, NJ, USA) FACS.

### 4.6. Transcriptomic Data Analysis

Available transcriptomic data from post-mortem lung tissues of COVID-19 patients and healthy controls were extracted from one previous study [28]. Differential analyses were performed with the DESeq2 package of R 4.0.3 software (R foundation, Ames, IA, USA) to compare the lung gene expression profiles between control (Lung_Ct_) and COVID-19 patients (Lung_Covid_). According to the authors, the genes with few reads or a False Discovery Rate (FDR) over 0.2 were removed by filtration [28].

### 4.7. Statistics

All the data are presented as mean ± SEM (standard error to the mean). Data from blood samples’ analysis, presented in the Table 1, were analysed using the R 4.0.3 software (R foundation, Ames, IA, USA) with a one-way ANOVA followed by a Tukey post-hoc, of which the *p*-values were corrected with the False Discovery Rate method (FDR). Discrete data (i.e., survival and co-morbidities of Table 1) were analysed with a general linear model test. AKR1B10 concentrations in the sera of patients presented in Figure 2 A,B were compared with a univariate Mann–Whitney test. All data obtained on cell lines were analysed in SigmaPlot 11.2 (Systat Software Inc., San-Jose, CA, USA) with a one-way ANOVA to determine the effects of inflammation (control vs. LPS), treatment (Control vs. Zopolrestat), or the AKR1B10_GFP_ protein in RAW264.7 and H1299 cells, if normally distributed. When significant, the omnibus test of variance was followed by a Holm–Sidak post-hoc. For non-parametric data, a box-plot transformation was performed to restore a parametric distribution, or a Tukey test was used to determine the nature of the effect.

## 5. Conclusions

In conclusion, this work highlights a critical role for AKR1B10 in COVID-19 disease that may participate to the cytokine storm, and proposes its inactivation as a potential contribution to the therapeutic arsenal to fight SARS-CoV2 Acute Respiratory Distress Syndrome and reduce the burden of the disease.

## Figures and Tables

**Figure 1 ijms-23-01911-f001:**
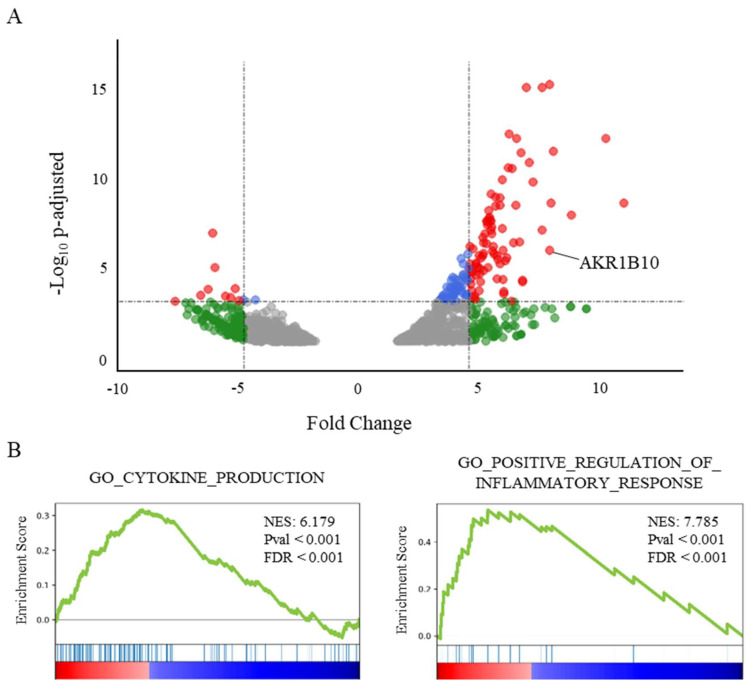
AKR1B10 is among the most frequently overexpressed genes in post-mortem lungs from severe COVID-19 patients. (**A**) Volcano plot showing the differential gene expression in lung tissue from deceased COVID-19 patients vs. healthy donors, reduced to genes over- or under-expressed with an FDR < 0.02 (from available transcriptomic data [28]). ● corresponds to an adjusted *p*-value < 0.001; ● corresponds to a fold change > 5; and ● corresponds to both. (**B**) GeneSet Enrichment Analysis (GSEA) plots of two genesets associated with the respective Gene Ontology terms CYTOKINE_PRODUCTION (NES: 6.18; FDR < 0.001) and POSITIVE_REGULATION_OF_INFLAMMATORY_RESPONSE (NES: 7.79; FDR < 0.001) illustrating two major components of the transcriptomic signature in lungs from severe COVID-19 patients.

**Figure 2 ijms-23-01911-f002:**
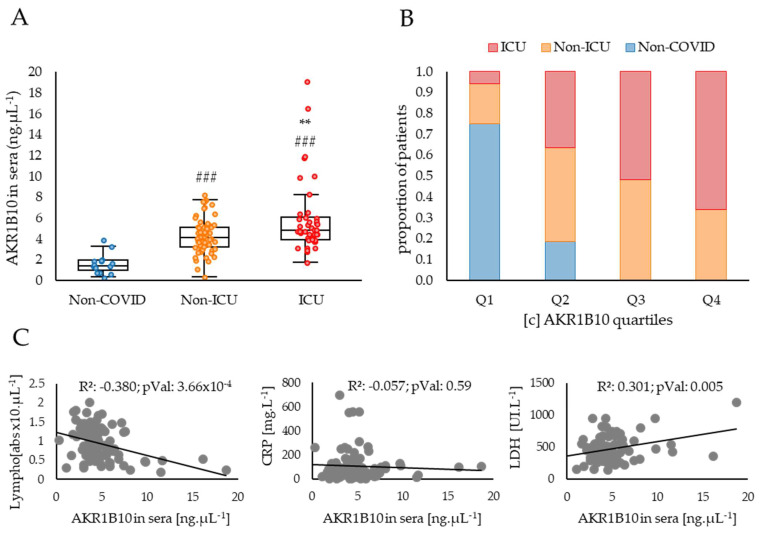
AKR1B10 concentration in sera is tightly associated with COVID-19 severity and correlated with other biological parameters known to be related to cytokine storm. (**A**) ELISA dosages of AKR1B10 in the blood of COVID-19 patients stratified into three groups corresponding to: “Non-COVID” (patients without COVID-19; *n* = 16, including 6 healthy individuals, 4 COPD and 6 cancer patients); “Non-ICU” (patients hospitalised in a non-ICU respiratory ward; *n* = 61), and “ICU” (patients admitted in an Intensive Care Unit; *n* = 43); (**B**) respective balanced proportions of patients of the non-COVID, non-ICU and ICU groups in each of the four quartiles (from Q1, the lowest, to Q4 the highest) of AKR1B10 sera concentrations; (**C**) correlations between AKR1B10 concentrations and Lymphocyte counts, CRP or LDH levels in the blood of COVID-19 patients. LDH: Lactate Dehydrogenase; Lympho: Lymphocyte counts; #: difference compared to Non-COVID individuals (###: *p* < 0.001); *: difference compared to non-ICU patients (**: *p* < 0.01).

**Figure 3 ijms-23-01911-f003:**
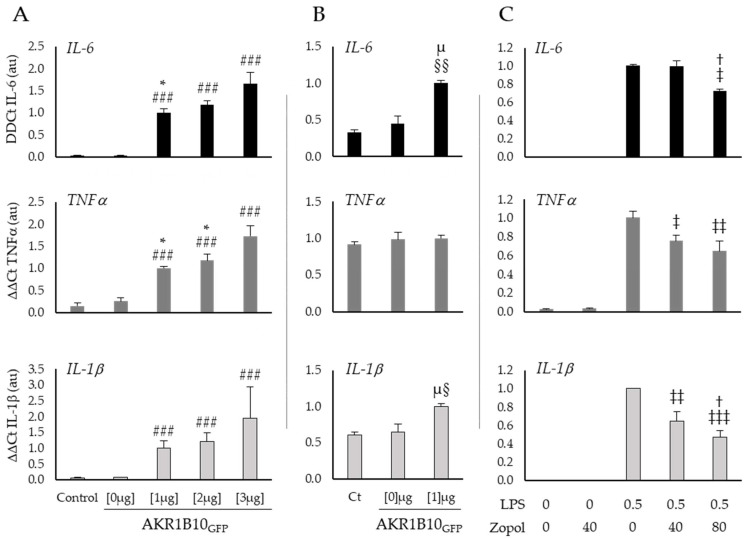
AKR1B10 is a key regulator of the cytokines production in RAW264.7 and H1299 cells, whose activity may be counteracted by pharmacological inhibitors. (**A**) Expression of the cytokines IL-6, TNFα and IL-1β, measured by RT-PCR in RAW264.7 macrophage cells after 12 h of 0 µg (Lipofectamine), 1 µg, 2 µg or 3 µg of peGFP-AKR1B10_GFP_ plasmid transfection; (**B**) expression of the cytokines IL-6, TNFα and IL-1β, measured by RT-PCR in lung cancer cells H1299 after 12 h of 0 (Lipofectamine) or 1 µg of peGFP-AKR1B10_GFP_ plasmid transfection; (**C**) effect of an AKR1B10 inhibitor (Zopolrestat at the indicated concentrations in mM) on cytokines expression in RAW264.7 cells stressed for 6 h by LPS, at the concentration of 0.5 µg·mL^−1^; LPS: Lipopolysaccharides; Zopol: Zopolrestat; (*n* = 3–5; mean ± SEM). #: difference compared to Control and pEGFP-AKR1B10_GFP_ [0 µg] ( ###: *p* < 0.001); *: difference compared to pEGFP-AKR1B10_GFP_ [3 µg] (*: *p* < 0.05); †: difference compared to LPS [0.5 µg·mL^−1^] and Zopolrestat [40 mM] (†: *p* < 0.05); §: difference compared to Control (§: *p* < 0.05; §§: *p* < 0.01); µ: difference compared to pEGFP-AKR1B10_GFP_ [0 µg] (µ: *p* < 0.05); ‡: difference compared to LPS [0.5 µg·mL^−1^] and Zopolrestat [0 mM] (‡: *p* < 0.05; ‡‡: *p* < 0.01; ‡‡‡: *p* < 0.001).

**Figure 4 ijms-23-01911-f004:**
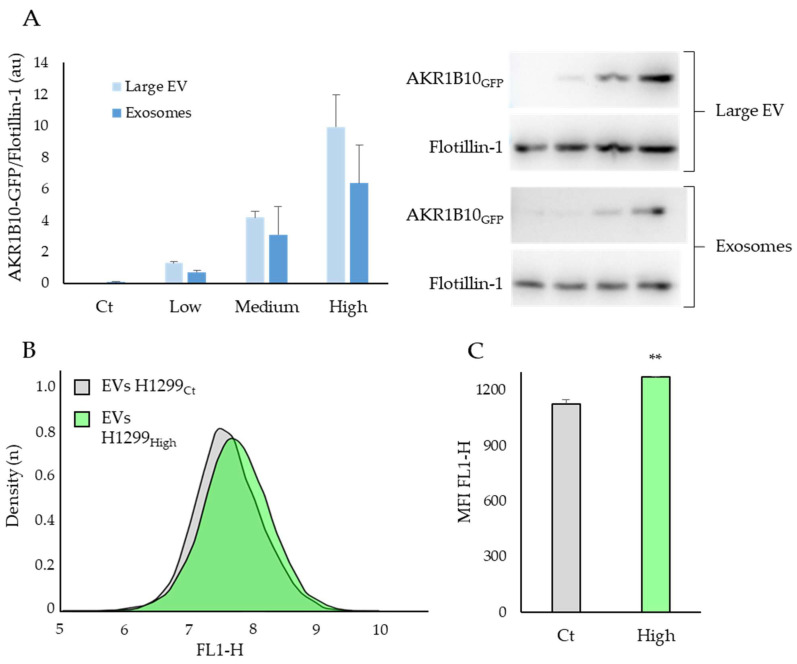
AKR1B10 can be transferred between different cells types via Extracellular Vesicles (EVs); (**A**) AKR1B10_GFP_ protein level in large EVs and exosomes extracted by centrifugation (Large EV: 10,000× *g* × 30 min; Exosomes: 100,000× *g* × 70 min) in the media of H1299 cells transfected with AKR1B10_GFP_ and sorted according to the terciles of GFP signal (respectively low, medium and high); (**B**) FACS measurements of GFP signal (FL1-H) of RAW264.7 cells exposed to extracellular vesicles extracted from H1299Ct and H1299High; (**C**) mean fluorescence intensity of the FL1-H signal measured by FACS (*n* = 3; mean ± SEM); EVs: Extracellular Vesicles. *: difference compared to EVs from AKR1B10_GFP_ Ct (**: *p* < 0.01).

**Table 1 ijms-23-01911-t001:** Characteristics of two groups of COVID-19 patients defined according to their hospitalisation either in a respiratory ward (*n* = 61) or in an intensive care unit (*n* = 43). (*n* = 104; mean ± SEM); ICU: Intensive Care Unit, BMI: Body Mass Index; PaO_2_: arterial pressure in oxygen; CT-Scan: percentage of lung ground glass opacity and area with more condensed aspect. HT: Hypertension; COPD: Chronic Obstructive Pulmonary Disease.

	Non-ICU			ICU			Adjusted *p*-Value
	Mean	±	SEM	Mean	±	SEM	
Sex (F/M)	26/35			18/25			
Survival (% [Surv-deceased])	98.4% [60/1]			37.2% [16/27]			<0.001
*Nb. of comorbidities:*							
0 (n (%))	39 (63.9%)			15 (34.9%)			<0.01
1 (n (%))	12 (19.7%)			16 (37.2%)			<0.05
>=2 (n (%))	10 (16.4%)			12 (27.9%)			0.16
*Nature of comorbidities:*							
Diabetes (n (%))	9 (15%)			13 (30%)			0.06
HT (n (%))	16 (26%)			15 (35%)			0.35
COPD (n (%))	2 (3%)			4 (9%)			0.20
Renal disease (n (%))	1 (2%)			2 (5%)			0.37
Cancer (n (%))	4 (7%)			7 (16%)			0.11
Age (year)	66.1	±	8.51	71.6	±	10.68	0.31
BMI (kg/m²)	26.4	±	3.66	26.8	±	4.12	0.31
PaO2 (mmHg)	59.9	±	7.84	53.3	±	8.21	<0.01
CRP (mg/L)	92.3	±	10.58	138.1	±	20.98	0.09
CT-Scan (%)	33.3	±	4.68	55.3	±	8.22	<0.01
Lymphocytes (abs. x10/µL)	1.57	±	0.17	0.86	±	0.13	<0.001
Fibrinogen (mg/dL)	570.4	±	73.54	588.5	±	87.59	0.99
D.Dimer (ng/mL)	2192	±	286.02	3860	±	560.52	0.31
Creatinine (mg/dL)	1.41	±	0.18	1.38	±	0.21	0.64
Ferritin (ng/mL)	546.0	±	55.65	917.1	±	152.55	<0.05
LDH (UI/I)	421.9	±	54.75	528.6	±	79.23	0.06
Procalcitonin (ng/mL)	0.32	±	0.04	1.01	±	0.63	0.31

## Data Availability

All important data is included in the manuscript.

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
