# Peer review of "AKR1B10, One of the Triggers of Cytokine Storm in SARS-CoV2 Severe Acute Respiratory Syndrome"

_ijms, 2022, doi:10.3390/ijms23031911_

Round 1
Reviewer 1 Report
The highlights are eye-catching and easy to understand. Not sure if they would be published eventually?
An interesting study that provides initial evidence for an association between AKR1B10 expression and severity of COVID-19. I would be hesistant to say it activates cytokine storm and acute systemic inflammation, as it appears to be activated in a variety of chronic inflammatory / cancer conditions as well.
Some typos in the table on P.5, "sexe", "lymphocytosis" (should be lymphocytes, given that you have a count, not T/F), "D.Dimeres" etc.
Please change Figure 2 Panel A to a boxplot (i.e. box and wisker plot). It is more appropriate than a bar chart here. The overlay with data points is okay. If a boxplot is used it can also replace Panel B. The sequence of plots in Panel C does not match that of the legend (lymphocytes, CRP, LDH vs LDH, "lymphocytosis, CRP).
Considering the "significant" association between AKR1B10 level and lymphocyte count / LDH (and potentialy other biochemical tests), is it possible to build a regression model to predict the serum AKR1B10 level in patients using these parameters? Arguably it would be clinically more pragmatic for hospitals that do not have the AKR1B10 assay, e.g. Clin Lab . 2015;61(9):1267-74. doi: 10.7754/clin.lab.2015.150203.
The authors may consider cross-reference to Front Pharmacol . 2020 Sep 15;11:584057. doi: 10.3389/fphar.2020.584057. eCollection 2020. to support their observation.
Author Response
- The highlights are eye-catching and easy to understand. Not sure if they would be published eventually?
We added the highlight sentences to the graphical abstract which will be published.
- An interesting study that provides initial evidence for an association between AKR1B10 expression and severity of COVID-19. I would be hesitant to say it activates cytokine storm and acute systemic inflammation, as it appears to be activated in a variety of chronic inflammatory / cancer conditions as well.
The referee is right. AKR1B10 is known to be activated in a series of chronic inflammatory diseases and cancers. However, taking into account that our data clearly confirm the published role of AKR1B10 in the activation of cytokines, we can assume that this activity of AKR1B10 is concentration-dependent and that the steady-state level of AKR1B10 in these chronic diseases is not compatible with the activation of the cytokine storm. Indeed, although we do not have a statistically significant number of cases, our non-COVID-19 sera samples composed of heathy donors (n = 6) COPD patients (n = 4), and lung cancer patients (n = 6) have lower AKR1B10 concentrations (Cf. Fig 2A). We also modified the text all along the document to avoid any misleading conclusion not based on our data or on the relevant literature (L45-46; L286-287, L425).
- Some typos in the table on P.5, "sexe", "lymphocytosis" (should be lymphocytes, given that you have a count, not T/F), "D.Dimeres" etc.
We apologize for these mistakes, a careful proofreading has been performed by a native English speaker.
- Please change Figure 2 Panel A to a boxplot (i.e. box and wisker plot). It is more appropriate than a bar chart here. The overlay with data points is okay. If a boxplot is used it can also replace Panel B. The sequence of plots in Panel C does not match that of the legend (lymphocytes, CRP, LDH vs LDH, "lymphocytosis, CRP).
Following the referee’s recommendations, we changed the bar chart by a boxplot with data points overlay. We choose to maintain Figure 2B since it visually represents the differences in the respective proportions of patients grouped according to their severity in the different AKR1B10 quartiles. We changed the Y-axis of the Figure 2C to match the legend.
- Considering the "significant" association between AKR1B10 level and lymphocyte count / LDH (and potentialy other biochemical tests), is it possible to build a regression model to predict the serum AKR1B10 level in patients using these parameters? Arguably it would be clinically more pragmatic for hospitals that do not have the AKR1B10 assay, e.g. Clin Lab . 2015;61(9):1267-74. doi: 10.7754/clin.lab.2015.150203.
We thank the referee for this suggestion. Following this remark, we performed a multiple regression model to predict AKR1B10 concentration based on the clinical parameters of the patients we have at our disposal. To determine which parameters allow the best prediction of AKR1B10, a step forward analysis was performed on the raw values or log10 of all available parameters. Based on this analysis, the best model includes Lymphocytose counts, Creatinine and LDH levels, and the age of the patients as parameters, but its predictive capacity is poor (AIC: 102.1; R²: 0.149) and it could not substitute a direct dosage of AKR1B10.
Figure Rev.1: Correlation between the AKR1B10 values obtained by measurement in sera of patients (Y-axis) and prediction by the statistic model performed with the step forward analysis (X-axis).
We understand that, currently, a dosage of AKR1B10 is not possible in the majority of hospitals but its growing interest in cancer diagnosis may quickly change this issue, especially thanks to new dosing methods such as the one cited by this referee.
- The authors may consider cross-reference to Front Pharmacol . 2020 Sep 15;11:584057. doi: 10.3389/fphar.2020.584057. eCollection 2020. to support their observation.
We thank this referee for bringing this reference to our attention. It is now cited and discussed in our discussion section (L313-316).
Reviewer 2 Report
In this work, Chabert and co-workers reported the positive correlation between sera AKR1B10 level to the severity of COVID-19 patients. They also showed that AKR1B10 overexpression can promote the expression of several cytokines: IL-6, TNFα, and IL-1β in RAW264.7 macrophage cell line and H1299 lung cancer cell line in vitro. Furthermore, they showed that AKR1B10 can be secreted in large extracellular vesicles and exosomes. RAW264.7 cells can receive AKR1B10 from extracellular vesicles secreted by H1299 lung cells, thus suggesting AKR1B10 might be a crucial systemic inflammatory regulator. Overall, AKR1B10 would be a potential biomarker evaluating the severity in COVID-19 patients, and AKR1B10 would be a potential therapeutic target in treating COVID-19-related cytokine storms. Although the data presented in this work looks convincing, the reviewer raises the following comments:
Major comments:
- According to the result in Figure 3C, the authors found that treating RAW 264.7 cells with lipopolysaccharides (LPS) could promote the cytokines (IL-6, TNFα, and IL-1β) expression, while further treating Zopolrestat, an AKR1B10 inhibitor can inhibit the cytokines expression. Can LPS treatment promote AKR1B10? To show the effect is AKR1B10 dependent, the reviewer highly recommends evaluating the effect of LPS treatment on AKR1B10 expression, and the effect of further Zopolrestat treatment on AKR1B10 expression.
- According to the result in Figure 4, the authors found that RAW264.7 cells can uptake AKR1B10 from AKR1B10 expressing H1299 cell extracellular vesicle. Will AKR1B10 containing extracellular vesicle/ exosome treated cells have higher expression of the investigated cytokines (e.g., IL-6, TNFα, and IL-1β)? This is particularly important to show whether the extracellular transportation of AKR1B10 is important for inducing inflammatory but not other confounding factors in this study.
Minor comments:
- The authors have mentioned that the prevention of the occurrence of severe COVID-19 forms is crucial to reduce the burden of the pandemic in the introduction session. It is also critical to discuss the recent advances in nanotechnologies for assisting the rapid and sensitive detection of SARS-CoV-2 for the screening of positively infected cases (e.g., J. Compos. Sci. 2021, 5(7), 190; ACS Appl. Mater. Interfaces 2022, 14, 3, 4714–4724)
- “All this parameters are known…” should be “All these parameters are known…”
- What is “LLC” in the left plot in Figure 2C? What does the unit stand for?
- In Figure 3C, Zopolrestat should be labelled on the x-axis, next to the “0 40 0 40 80”
- In the graphic abstract, the infected person should be labelled as the patient with COVID-19 and the symptoms of the cytokine storm.
- The terminology difference between EVs and exosomes should be defined (e.g., size, content, etc. Relevant references can be found in Theranostics. 2022; 12(1): 207–231).
- There are typoes “Sars-CoV-2” instead of “SARS-CoV-2” in the main text.
- The sentence “Thus, the increased production of TNFα, IL-6, and IL-β observed….” Where there is an extra space between “and” and “IL- β”. The authors should carefully check these typoes.
- The authors should further check the grammatical mistakes throughout the manuscript.
Author Response
In this work, Chabert and co-workers reported the positive correlation between sera AKR1B10 level to the severity of COVID-19 patients. They also showed that AKR1B10 overexpression can promote the expression of several cytokines: IL-6, TNFα, and IL-1β in RAW264.7 macrophage cell line and H1299 lung cancer cell line in vitro. Furthermore, they showed that AKR1B10 can be secreted in large extracellular vesicles and exosomes. RAW264.7 cells can receive AKR1B10 from extracellular vesicles secreted by H1299 lung cells, thus suggesting AKR1B10 might be a crucial systemic inflammatory regulator. Overall, AKR1B10 would be a potential biomarker evaluating the severity in COVID-19 patients, and AKR1B10 would be a potential therapeutic target in treating COVID-19-related cytokine storms. Although the data presented in this work looks convincing, the reviewer raises the following comments:
Major comments:
- According to the result in Figure 3C, the authors found that treating RAW 264.7 cells with lipopolysaccharides (LPS) could promote the cytokines (IL-6, TNFα, and IL-1β) expression, while further treating Zopolrestat, an AKR1B10 inhibitor can inhibit the cytokines expression. Can LPS treatment promote AKR1B10? To show the effect is AKR1B10 dependent, the reviewer highly recommends evaluating the effect of LPS treatment on AKR1B10 expression, and the effect of further Zopolrestat treatment on AKR1B10 expression.
Following this reviewer’s suggestion, we tested the effect of a LPS treatment on AKR1B10 expression. The resulting data show that, in RAW 264.7 and H1299 cells, an exposure to LPS during 6h at 1µg.ml-1 does not induce an increase in AKR1B10 expression (Cf. Suppl. Fig S2B). This observation is in line with the current literature which does not report any significant change of AKR1B10 expression in transcriptomic data of RAW264.7 cells exposed to different lipopolysaccharides for 6h at 50ng.ml-1 [1]. Nonetheless, at a higher concentration of LPS (2 and 3µg.ml-1) we observe increased AKR1B10 concentrations (not shown), suggesting that the contribution of AKR1B10 on cytokines expression, observed in the present study and in the literature, is driven by both the basal activity of this enzyme in the cells, as well as an increase of the protein in the context of high cellular stress. We added a sentence in the conclusion (L305-309).
- Rutledge, H.R.; Jiang, W.; Yang, J.; Warg, L.A.; Schwartz, D.A.; Pisetsky, D.S.; Yang, I.V. Gene Expression Profiles of RAW264.7 Macrophages Stimulated with Preparations of LPS Differing in Isolation and Purity. Innate Immun 2012, 18, 80–88, doi:10.1177/1753425910393540.
According to the result in Figure 4, the authors found that RAW264.7 cells can uptake AKR1B10 from AKR1B10 expressing H1299 cell extracellular vesicle. Will AKR1B10 containing extracellular vesicle/ exosome treated cells have higher expression of the investigated cytokines (e.g., IL-6, TNFα, and IL-1β)? This is particularly important to show whether the extracellular transportation of AKR1B10 is important for inducing inflammatory but not other confounding factors in this study.
We agree with the reviewer’s comment, that showing an increased production of in cells exposed to extracellular vesicles would have been a clear way to demonstrate the association between AKR1B10 transfer and inflammation in distant cells. Nonetheless, we set up this in vitro system to demonstrate that these AKR1B10-containing vesicles are able to deliver the protein in “naïve” cells, but this experimental setting is not appropriate to test the induction of cytokines expression. Indeed, in our system, the AKR1B10 concentrations measured in the cell media are far from reaching the concentrations measured in our series of COVID-19 patients (Cf. Suppl. Fig. S2A), even after EVs extraction. Therefore, although this system is sensitive enough to demonstrate the presence of AKR1B10 vesicles capable of fusing to other cells, it does not allow to reach AKR1B10 concentrations in similar ranges as measured in the patients’ serum and it is not able to test the induction of cytokines expression. For all these reasons, the conclusions concerning this part have been downtoned in the text (they are now in conditional tense; L330-337). We also added a short paragraph in the strength and weakness section to highlight this point (L347-353).
Minor comments:
- The authors have mentioned that the prevention of the occurrence of severe COVID-19 forms is crucial to reduce the burden of the pandemic in the introduction session. It is also critical to discuss the recent advances in nanotechnologies for assisting the rapid and sensitive detection of SARS-CoV-2 for the screening of positively infected cases (e.g., J. Compos. Sci. 2021, 5(7), 190; ACS Appl. Mater. Interfaces 2022, 14, 3, 4714–4724)
We thank the referee for bringing this important point to our attention. This point is now discussed in the introduction and the corresponding reference is now cited (L57).
- “All this parameters are known…” should be “All these parameters are known…”
We corrected the sentence and carefully proofread the manuscript to remove grammatical mistakes.
- What is “LLC” in the left plot in Figure 2C? What does the unit stand for?
We changed the Y-axes legends of Figure 2C by “Lympho (abs. x10/µL)” and explained the abbreviation in the legend of the figure.
- In Figure 3C, Zopolrestat should be labelled on the x-axis, next to the “0 40 0 40 80”
We apologize for the mistake and corrected the figure according to the reviewer' comment.
- In the graphic abstract, the infected person should be labelled as the patient with COVID-19 and the symptoms of the cytokine storm.
Following the reviewer’s suggestion, we added “COVID-19 patient” above the silhouette of the graphical abstract.
- The terminology difference between EVs and exosomes should be defined (e.g., size, content, etc. Relevant references can be found in Theranostics. 2022; 12(1): 207–231).
We added a short sentence to describe the difference between the EVs sub-groups used in the study and cited the article suggested by the reviewer (L251-254).
- There are typoes “Sars-CoV-2” instead of “SARS-CoV-2” in the main text.
We thank the reviewer for this comment and carefully proofread the article to remove mistakes.
- The sentence “Thus, the increased production of TNFα, IL-6, and IL-β observed….” Where there is an extra space between “and” and “IL- β”. The authors should carefully check these typoes.
These typoes were corrected.
- The authors should further check the grammatical mistakes throughout the manuscript.
We apologize for these mistakes, and the whole manuscript has been carefully proofread.
Round 2
Reviewer 2 Report
The authors appropriately addressed the issues raised by the reviewer.